# Single-cell and single-nucleus RNA-sequencing from paired normal-adenocarcinoma lung samples provide both common and discordant biological insights

**Sébastien Renaut**[1], **Victoria Saavedra Armero**[1], **Dominique K. Boudreau**[1], **Nathalie Gaudreault**[1], **Patrice Desmeules**[1], **Sébastien Thériault**[1], **Patrick Mathieu**[1], **Philippe Joubert**[1], **Yohan Bossé**[1,2]*

1 Institut universitaire de cardiologie et de pneumologie de Québec–Université Laval, Quebec City, Canada,
2 Department of Molecular Medicine, Université Laval, Quebec City, Canada

* yohan.bosse@criucpq.ulaval.ca

## Abstract

Whether single-cell RNA-sequencing (scRNA-seq) captures the same biological information as single-nucleus RNA-sequencing (snRNA-seq) remains uncertain and likely to be context-dependent. Herein, a head-to-head comparison was performed in matched normal-adenocarcinoma human lung samples to assess biological insights derived from scRNA-seq versus snRNA-seq and better understand the cellular transition that occurs from normal to tumoral tissue. Here, the transcriptome of 160,621 cells/nuclei was obtained. In non-tumor lung, cell type proportions varied widely between scRNA-seq and snRNA-seq with a predominance of immune cells in the former (81.5%) and epithelial cells (69.9%) in the later. Similar results were observed in adenocarcinomas, in addition to an overall increase in cell type heterogeneity and a greater prevalence of copy number variants in cells of epithelial origin, which suggests malignant assignment. The cell type transition that occurs from normal lung tissue to adenocarcinoma was not always concordant whether cells or nuclei were examined. As expected, large differential expression of the whole-cell and nuclear transcriptome was observed, but cell-type specific changes of paired normal and tumor lung samples revealed a set of common genes in the cells and nuclei involved in cancer-related pathways. In addition, we showed that the ligand-receptor interactome landscape of lung adenocarcinoma was largely different whether cells or nuclei were evaluated. Immune cell depletion in fresh specimens partly mitigated the difference in cell type composition observed between cells and nuclei. However, the extra manipulations affected cell viability and amplified the transcriptional signatures associated with stress responses. In conclusion, research applications focussing on mapping the immune landscape of lung adenocarcinoma benefit from scRNA-seq in fresh samples, whereas snRNA-seq of frozen samples provide a low-cost alternative to profile more epithelial and cancer cells, and yield cell type proportions that more closely match tissue content.

**Data Availability Statement:** The datasets generated by Cellranger are available as open-

access downloadable files on Zenodo (10.5281/zenodo.11205626). All analytical codes used to produce the results of this study are available at https://github.com/Yohan-Bosse-Lab/scRNA.

**Funding:** This work was supported by the IUCPQ Foundation and a generous donation from Mr. Normand Lord. P.M. is the recipient of the Joseph C. Edwards Foundation granted to Université Laval. P.J. is the recipient of a Junior 2 Clinical Research Scholar award from the Fonds de recherche Québec - Santé (FRQS). Y.B. holds a Canada Research Chair in Genomics of Heart and Lung Diseases. The funders had no role in study design, data collection and analysis, decision to publish, or preparation of the manuscript.

**Competing interests:** The authors have declared that no competing interests exist.

## Author summary

Single-cell transcriptomic datasets provide unprecedented opportunities to disentangle the complex tissue microenvironment and cellular origin of cancer. Data are scarce regarding the pros and cons of single-cell RNA sequencing (scRNA-seq) of freshly explanted human tissues over single-nuclei sequencing (snRNA-seq) from the same archived frozen tissues. Lung adenocarcinoma represents a medically valuable case study to compare the biological signal recovered through cells and nuclei sequencing. Here, we sequenced the transcriptome of 160,621 cells/nuclei in paired normal-adenocarcinoma lung samples. Cell type proportions varied widely between scRNA-seq and snRNA-seq with a predominance of immune cells in the former and epithelial cells in the later. Adenocarcinomas were characterized by an increase in cell type heterogeneity and a greater prevalence of malignant epithelial cells in both scRNA-seq and snRNA-seq. The cellular and gene expression transition that occur from normal lung to adenocarcinoma showed common and discordant biological insights whether cells or nuclei were examined. Research applications focussing on mapping the immune landscape of lung cancer benefit from scRNA-seq in fresh samples, whereas snRNA-seq of the same frozen samples provide a low-cost and more flexible alternative to profile more epithelial and cancer cells, and yield cell type proportions that more closely match tissue content.

## Introduction

Single-cell sequencing (scRNA-seq) has the ability to inspect the cellular heterogeneity of tissue and cancer with unprecedented details, and as such provides important insights into the cellular origin and cell-specific molecular defects that play a role in disease pathogenesis [1–4]. However, given the pace at which the field is evolving, uncertainties remain with respect to the design and analysis of single-cell transcriptomic datasets in order to gain the most from biological samples. Fresh biospecimens are generally prioritized for cell viability and greater yield of high-quality cells. For tissues, scRNA-seq requires disaggregating the tissue to release individual cells into a single-cell suspension. Differences in dissociation and sample preparation efficiency across cell types are known to affect RNA integrity and can skew cell type proportions. A well-known instance of dissociation bias is observed in human lung tissue, where dissociation of fresh tumor (biopsies or resected specimens) commonly results in a majority of immune cells being sequenced [5–7]. While the aforementioned cell-type dissociation bias can be partly alleviated by enriching the epithelial cell fraction using EPCAM-based cell sorting [6], single cell preparation protocols may also affect cell viability and introduce transcriptional signatures associated with dissociation and stress responses [6,8,9].

Analyzing nuclei (single-nucleus sequencing or snRNA-seq) instead of cells has been proposed as an alternative for frozen samples and tissues that cannot be readily dissociated [10,11]. While cellular compositions recovered from scRNA-seq versus snRNA-seq can vary substantially [12], the transition from cell to nucleus sequencing may help to reduce the dissociation bias and transcriptional stress responses, facilitate the study of difficult-to-dissociate tissues and cell types, and allow the assessment of large cells that cannot pass through micro-fluidics systems. At the same time, reference databases and cell type-specific gene markers, which are readily used to annotate unknown cell populations, have been largely built from scRNA-seq datasets [4] and therefore may not be optimal for snRNA-seq. Cell types and gene expression differences between scRNA-seq and snRNA-seq have been observed in mouse

kidneys [13,14] and brain [15,16] as well as in human metastatic breast cancer and neuroblastoma [12]. Combining scRNA-seq and snRNA-seq technologies from matched samples has been shown to better capture cell heterogeneity and produce a more comprehensive cell map of healthy human liver [17]. However, head-to-head comparisons between scRNA-seq and snRNA-seq are still scarce and to the best of our knowledge, this direct comparison has never been evaluated in the context of patient-matched normal lung and tumor tissues.

Lung cancer is highly prevalent and the number one cause of cancer mortality. It thus represents a medically valuable case study to compare the biological signal recovered through cells and nuclei sequencing. A variety of experimental designs and samples have been evaluated by scRNA-seq in patients with lung cancer. This includes lung samples enriched (e.g. FACS-sorted) for immune cells [18,19], lung tumor of mixed histological types [2,7], and non-small cell lung cancer (NSCLC) samples before and after targeted therapy [20] or immunotherapy [21]. More specifically in lung adenocarcinomas (LUAD), the most common histological subtype of lung cancer, which originates from epithelial cells that line the inside of the lungs, resected specimens or biopsies from two to eleven [2,5–7,22] patients have been evaluated, but with a very limited number of paired normal-adenocarcinoma lung samples. Compared with normal lung samples, epithelial cells from lung adenocarcinomas were characterized by a depletion of alveolar cells (AT1 and AT2) [2,6], lost cell identity and more cells annotated as mixed-lineage [5,23], higher transcriptome complexity and cell heterogeneity [6,24], patient-specific cancer cell clusters [20,25], transcriptional states associated with survival [22,23], and AT2 cells dedifferentiated into a stem-like state [24] or alveolar intermediary cells that could act as progenitors of *KRAS*-driven LUAD [25].The shift in immune cells from normal to LUAD samples observed in previous studies were similarly informative. It unveiled an increase in B, plasma and T regulatory cells coupled with a decline in natural killer cells as well as reduced signatures of cytotoxicity in T cells, antigen presentation in macrophages, and inflammation in dendritic cells, which are all coherent features of an immunosuppressive tumor microenvironment [6,18]. Finally, differentially enriched ligand-receptor interactions promoting tumorigenesis were also observed between LUADs and normal tissues [6,22].

Herein, specimens derived from the same patients were tested using both scRNA-seq in fresh tissues and snRNA-seq from flash frozen tissues using the 10x Genomics workflow. The biology captured by both methods was compared in the context of paired tumor-normal human lung samples explanted from patients that underwent surgery for lung adenocarcinoma. This study design revealed the cellular and molecular transitions that occur from normal lung to adenocarcinoma, and evaluated the commonality and discordance in the stemming biological insights gained from cells versus nuclei. In addition, we compared the same paired normal-adenocarcinoma human lung samples using an immune cell depletion protocol that alleviates the cell-type dissociation bias, with the aim of recovering a more representative biological signal.

## Results

### Experimental design

Four patients, two tissue types (Normal/Tumor) and three experimental methods (scRNA-seq, snRNA-seq & immune-depleted scRNA-seq, hereafter labelled as *Cell*, *Nucleus* and *Immune-depleted cell*) were processed for a total of twenty-four samples. The experimental design is presented in **Fig 1**. The four patients underwent lung cancer surgery with pathologically confirmed LUAD (**Fig 1A**). The clinical characteristics of patients are detailed in **Table A in S1 Text**. Both LUAD and normal lung specimens were obtained from each patient (**Fig 1B**). Fresh tissues were immediately processed for scRNA-seq and adjacent samples were flashed

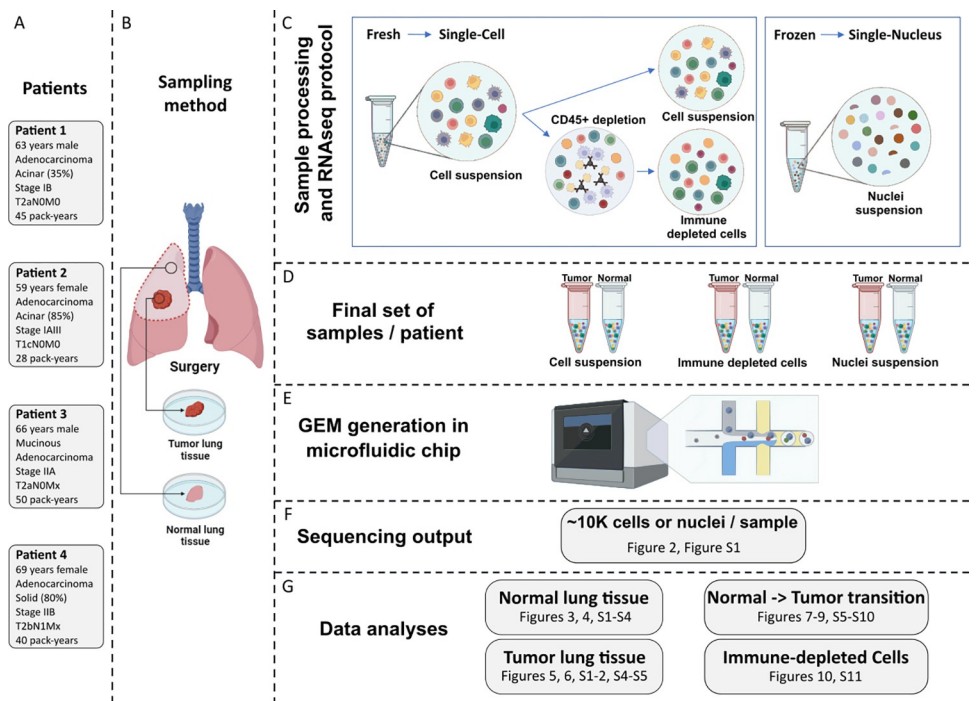

**Fig 1. Overview of the experimental design.** For each patient (**A**), a Tumor specimen and a Normal (non-malignant) lung specimen harvested from a site distant from the tumor were resected (**B**). The research specimens were immediately divided into smaller fragments. For both Normal and Tumor lung specimens, a fragment was frozen in liquid nitrogen and stored at -80˚C until further processing for snRNA-seq. For fresh specimens, the fragments proceeded directly to dissociation into single-cell suspensions. A subsample of the dissociation mix underwent immune cell depletion (**C**). The final set of samples (**D**) were then loaded in wells of the microfluidic chip (**E**) in order to generate the transcriptome of approximately 10,000 cells or nuclei per sample (**F**). Dataset comparisons performed with accompanying figures (**G**). Images created with Biorender.

frozen and stored at -80˚C until further processing for snRNA-seq (**Fig 1C**). The single cell suspensions dissociated from fresh tissues were also submitted to CD45+ immune depletion, leading to three cell suspensions per specimen and thus six per patient (**Fig 1D**). The characteristics of samples and cell/nucleus suspensions are presented in **Table B in S1 Text**. Single cell suspensions were converted to libraries using the 10x Genomics workflow (**Fig 1E**) and sequenced on an Illumina NextSeq 2000 aiming for ~10,000 cells or nuclei per sample (**Fig 1F**). We partitioned the analysis by focusing on 1) normal lung tissues, 2) LUAD tissues, 3) paired normal-adenocarcinoma lung samples, and 4) immune-depleted samples (**Fig 1G**).

## Overview of the dataset

A total of 160,621 cells/nuclei passed quality control (53,286; 57,078 and 50,257 for *Cell*, *Nucleus* and *Immune-depleted cell* datasets respectively). Uniform manifold approximation and projections (UMAP) of all cells coloured by cell types, tissue types, experimental methods and patients are provided in **S1 Fig**. On average, we observed 6,692 cells per sample (6,661; 7,135 and 6,282 for *Cell*, *Nucleus* and *Immune-depleted cell* datasets respectively, **Fig 2A**) and detected 2,216 genes per cell (1,868; 2,309 and 2,473 genes for *Cell*, *Nucleus* and *Immune-depleted-cell* datasets respectively, **Fig 2B**).

From the 61 finest cell types annotations defined by the Human Lung Cell Atlas (HLCA) [4], 35 were present in the current dataset at a frequency of >100 cells and we were able to annotate confidently 97.7% of cells at the coarsest level (*immune*, *epithelial*, *endothelial*,

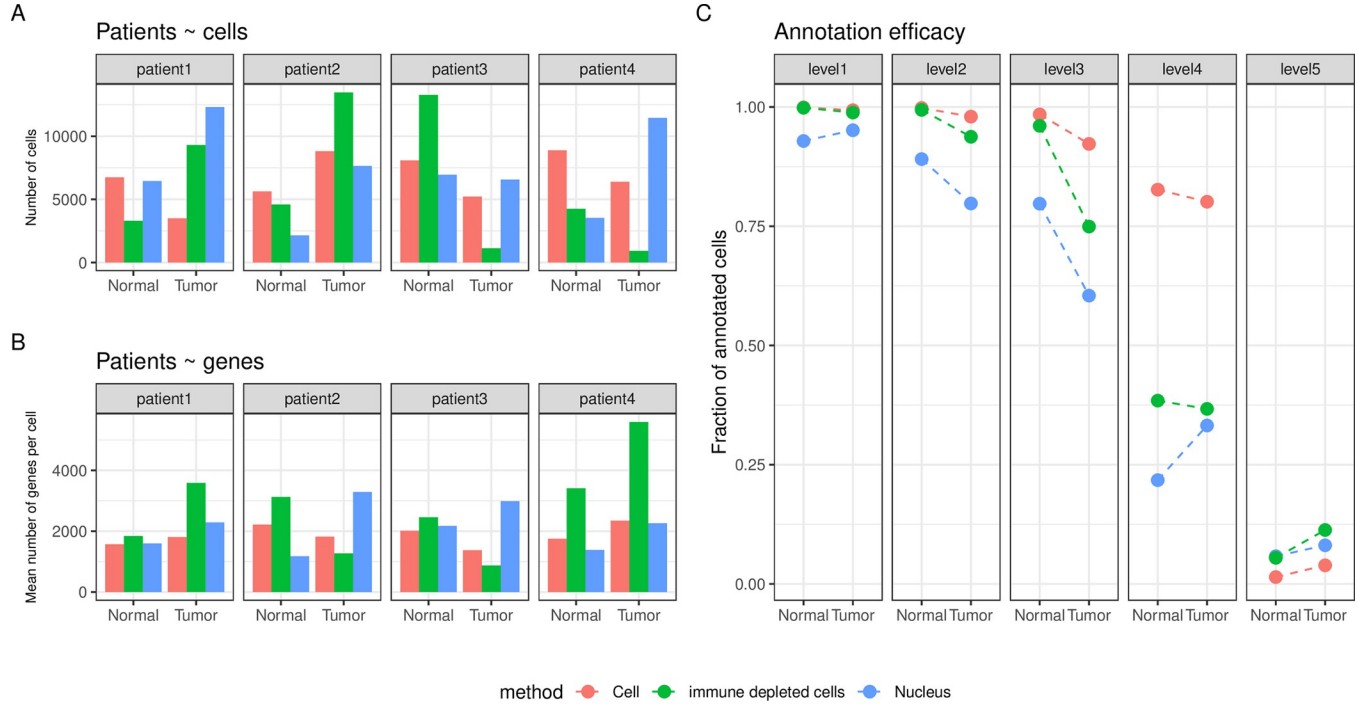

**Fig 2. Overview of the 160,621 cells/nuclei that passed quality control obtained from lung Tumors and distal Normal lung samples. A.** Number of cells retained after quality control for each patient, each experimental method (*Cell*, *Nucleus*, *Immune-depleted cell*) and tissue type (Normal, Tumor). **B.** Mean number of genes per cell, per patient, method and tissue type. **C.** The fraction of annotated cells for each of the five-level HLCA hierarchical cell annotation reference framework, per method and tissue type.

*stroma*, **Fig 2C** and **Table C in S1 Text**). This reference-based mapping and annotation approach is consistent with a marker-based approach for both the *Cell* and *Nucleus* datasets (**S2 Fig**). Nevertheless, cell type annotation scores were significantly lower (smaller fraction of annotated cells) in the *Nucleus* compared to the *Cell* dataset (two-way ANOVA, *p*-value < 2e-16), fine-level compared to high-level annotations (*p*-value < 2e-16) and Tumor compared to Normal tissue (*p*-value < 2e-16).

## scRNA and snRNA of Normal lung tissue

In **Fig 3**, the UMAP visualisation showed that the *Cell* dataset from Normal lung tissue was largely dominated by immune cells, with 23,044 immune cells (81.5% of total, **Fig 3A**). Conversely, the *Nucleus* dataset was dominated by epithelial cells, with 12,556 epithelial cells (69.9%, **Fig 3B**). In addition, the *Nucleus* dataset contained a larger fraction of unclassified cells compared to the *Cell* dataset (7.3% vs 0.1%, Fisher Exact Test [FET], *p*-value < 2e-16). These results were consistent across individual patients (**S3 Fig**).

As expected, on histologic evaluation, the proportions of epithelial and immune cells were consistent with the *Nucleus*, rather than the *Cell* dataset **S4A and S4B Fig**).

To further refine the immune community of cells, we subsetted only the immune cells and labelled the plots with a finer level (level 3) annotation (*Cell*, **Fig 3C**; *Nucleus*, **Fig 3D**). We observed that the *Cell* dataset provided a better fine-grained classification as proportionally more cells could be classified into specific cell types. To this effect, the *Nucleus* dataset contained a larger fraction of unclassified cells (41.7% vs 0.7%, FET, *p*-value < 2e-16).

We repeated this subsetting approach for epithelial cells, given their primary role in the onset of lung adenocarcinoma. We observed that *Cell* samples form distinct clusters mainly

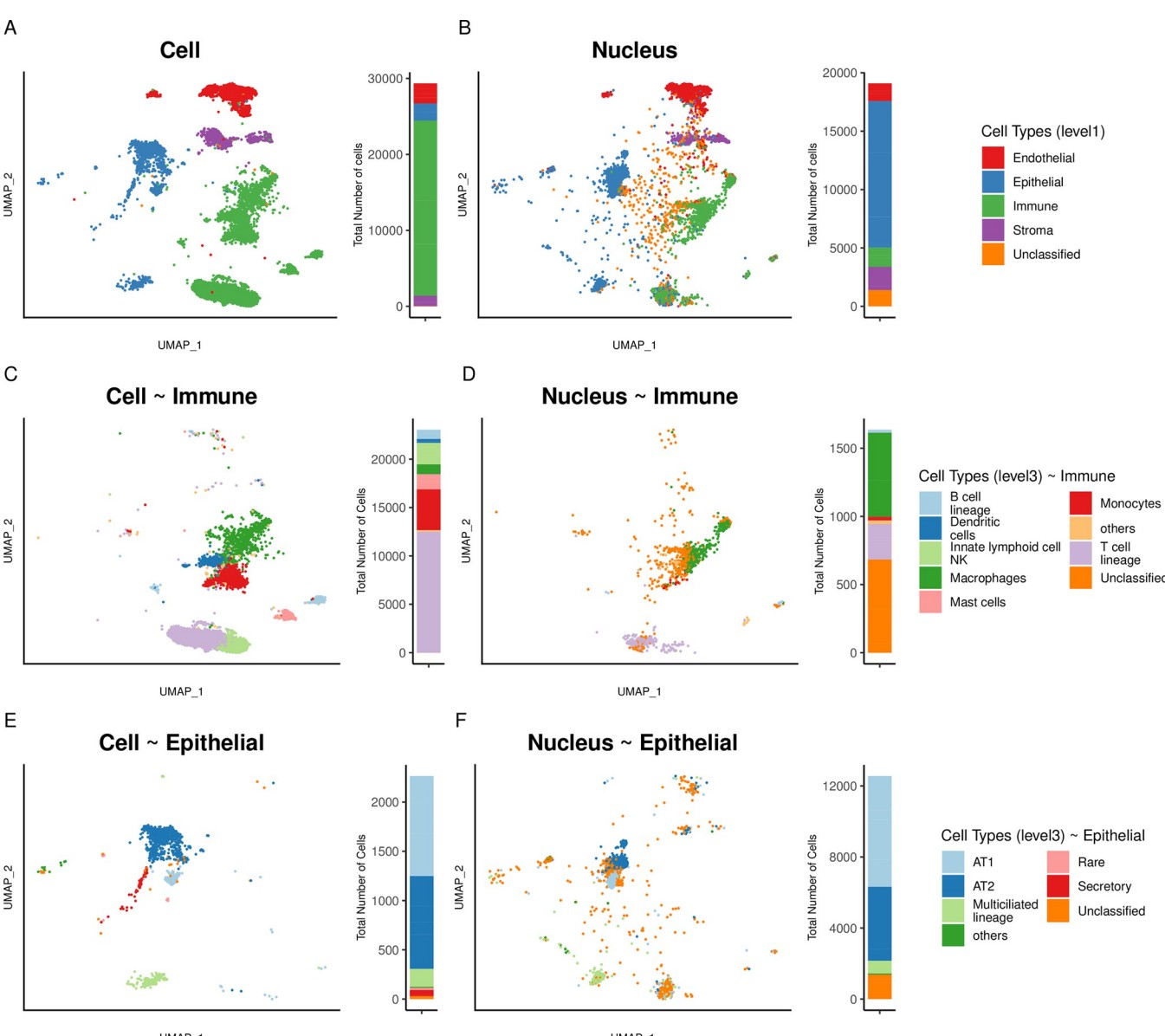

**Fig 3. UMAP representations and cell types annotations of Normal lung tissue** for *Cell* (**A**) and *Nucleus* (**B**) datasets with general cell types (level 1) annotation. Finer-grained annotation (level 3) for the subset of immune cells (**C**) or nuclei (**D**) and for the subset of epithelial cells (**E**) or nuclei (**F**). To the right of each UMAP, stacked bar plots indicate the proportion of each cell type in the specific dataset. Cell types present at < 1% are labelled as others.

composed of AT1, AT2 and multiciliated lineages (**Fig 3E**). The *Nucleus* dataset, which had more than five times more epithelial cells than the *Cell* dataset (12,556 versus 2,264), contained similar cell types and mainly in similar proportions, except for a sizable fraction of unclassified cells that appeared largely scattered in the UMAPs (10.9% unclassified in *Nucleus* versus 1.29% in *Cell*, FET, *p*-value < 2e-16, **Fig 3F**).

In **Fig 4**, we present, for each cell type (level 3 annotation), the fraction of cells originating from each patient (**Fig 4A**), the number of cells (**Fig 4B**) and the number of genes per cell (**Fig 4C**). In **Fig 4D–4F**, we present the same information for the *Nucleus* dataset and this visualization confirmed that the *Nucleus* dataset has similar cellular composition, except for the over-representation of immune cells in the *Cell* dataset. Both in *Cell* and *Nucleus* datasets,

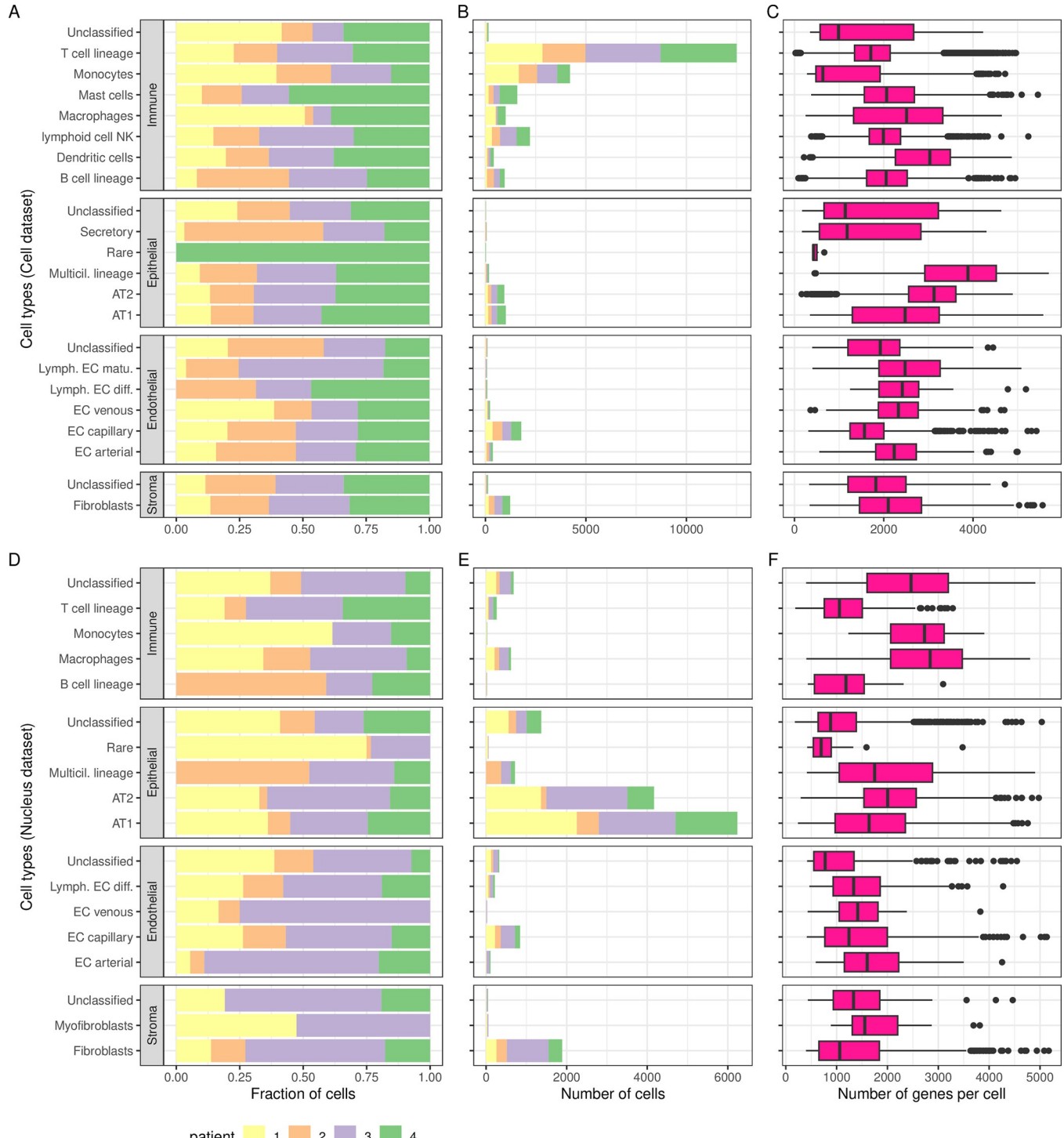

**Fig 4. Cell types characteristics of Normal lung tissue.** For each of the four coarse (level 1) cell types annotation (*Immune*, *Epithelial*, *Endothelial*, *Stroma*) further refined into finer categories (level 3), the fraction of cells (**A**: *Cell* dataset, **D**: *Nucleus*) and the number of cells (**B**: *Cell*, **E**: *Nucleus*) originating from each patient. Box plots of the number of genes expressed per cell (**C**: *Cell*, **F**: *Nucleus*), with plot center, box and whiskers corresponding to median, IQR and 1.5 × IQR, respectively. Note that only cell types with > 20 cells were retained for clarity in this visual representation.

epithelial cell types were dominated by AT1 first and then AT2; endothelial cell types were dominated by capillary cells; and stromal cell types were dominated by fibroblasts. With respect to the number of genes (transcripts) per cell (**Fig 4C and 4F**), we observed many discordant patterns between *Nucleus* and *Cell* datasets, indicating that similar cell types presented different overall transcriptional signatures based on the experimental method. For example, in the *Cell* dataset, median numbers of genes per cell were low for monocytes (635), but high for T cells (1,709), and the pattern was in the opposite direction for the *Nucleus* dataset (Monocytes = 2,729, T cells = 1,055). For their part, alveolar cells AT1 and AT2 contained 50% more genes expressed in the *Cell* dataset (AT1: 2,479 and AT2: 3,126) compared to the *Nucleus* (AT1: 1,639 and AT2: 2,004), and fibroblast two times as much (2,101 vs 1,061).

## scRNA and snRNA of LUAD

In **Fig 5A**, the UMAPs showed that *Cell* sequencing samples from lung Tumor tissues were largely dominated by immune cell types (20,410 immune cells vs 5,764 in *Nucleus* dataset), while in **Fig 5B**, the *Nucleus* dataset were dominated by epithelial cells (27,362 epithelial cells in *Nucleus* vs 1,220 in *Cell* dataset). The predominance of immune cells in *Cell* and epithelial cells in *Nucleus* were observed across the four patients (**S5 Fig**). The *Nucleus* showing again a more accurate reflection of the real cellular composition of LUAD assessed by immunohistochemical staining (**S4A and S4B Fig**).

For both *Cell* and *Nucleus* datasets, cells appeared more scattered (i.e., more heterogeneous) in the Tumor compared to Normal lung (median Silhouette index $_{(Normal)}$ = 0.69; median Silhouette index $_{(Tumor)}$ = 0.53; two-way ANOVA, *p*-value $< 2e\text{-}16$, **S6 Fig**). This shows a suboptimal cell type assignment of Tumor samples to the described lung cell types from the HLCA reference.

In **Fig 6**, we present, for each level 3 annotation cell type, the fraction of cells from each patient (**Fig 6A**), the number of cells (**Fig 6B**), and the number of genes per cell (**Fig 6C**). In **Fig 6D–6F**, we present the same information for the *Nucleus* dataset. First, we observed, within a coarse level annotation, similar cell types and similar proportions in *Cell* and *Nucleus* datasets. For example, T cells largely dominated the immune cells, fibroblasts dominated the

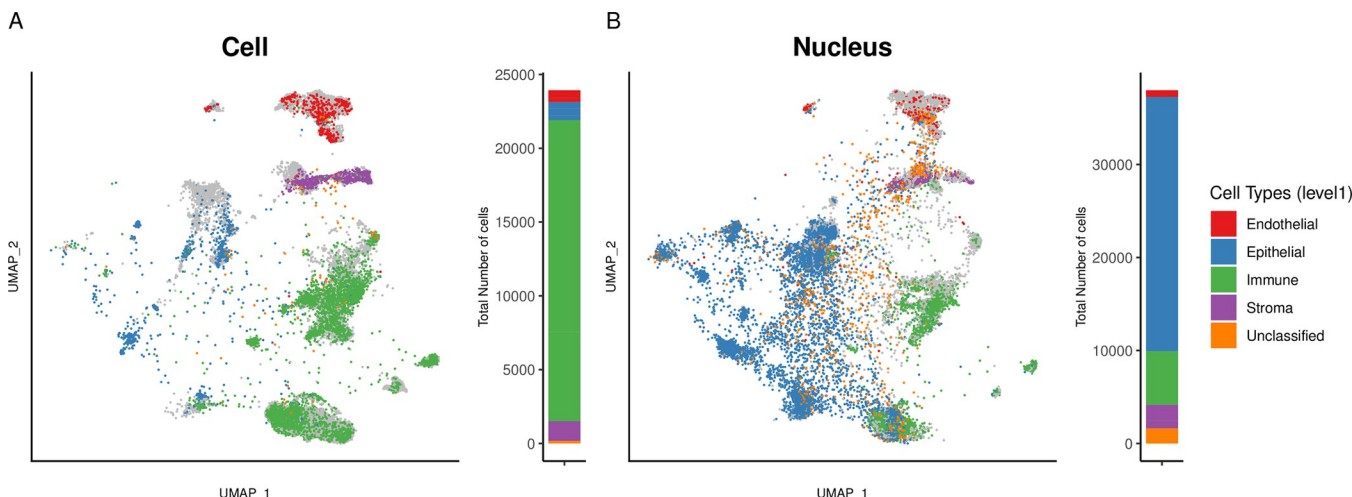

**Fig 5. UMAP representations and cell types annotations of Tumor tissue** for *Cell* (**A**) and *Nucleus* (**B**) datasets with general cell types (level 1) annotation. Tumor samples are overlaid on top of Normal samples (in gray). To the right of each UMAP, stacked bar plots indicate the proportion of each cell type in the specific dataset.

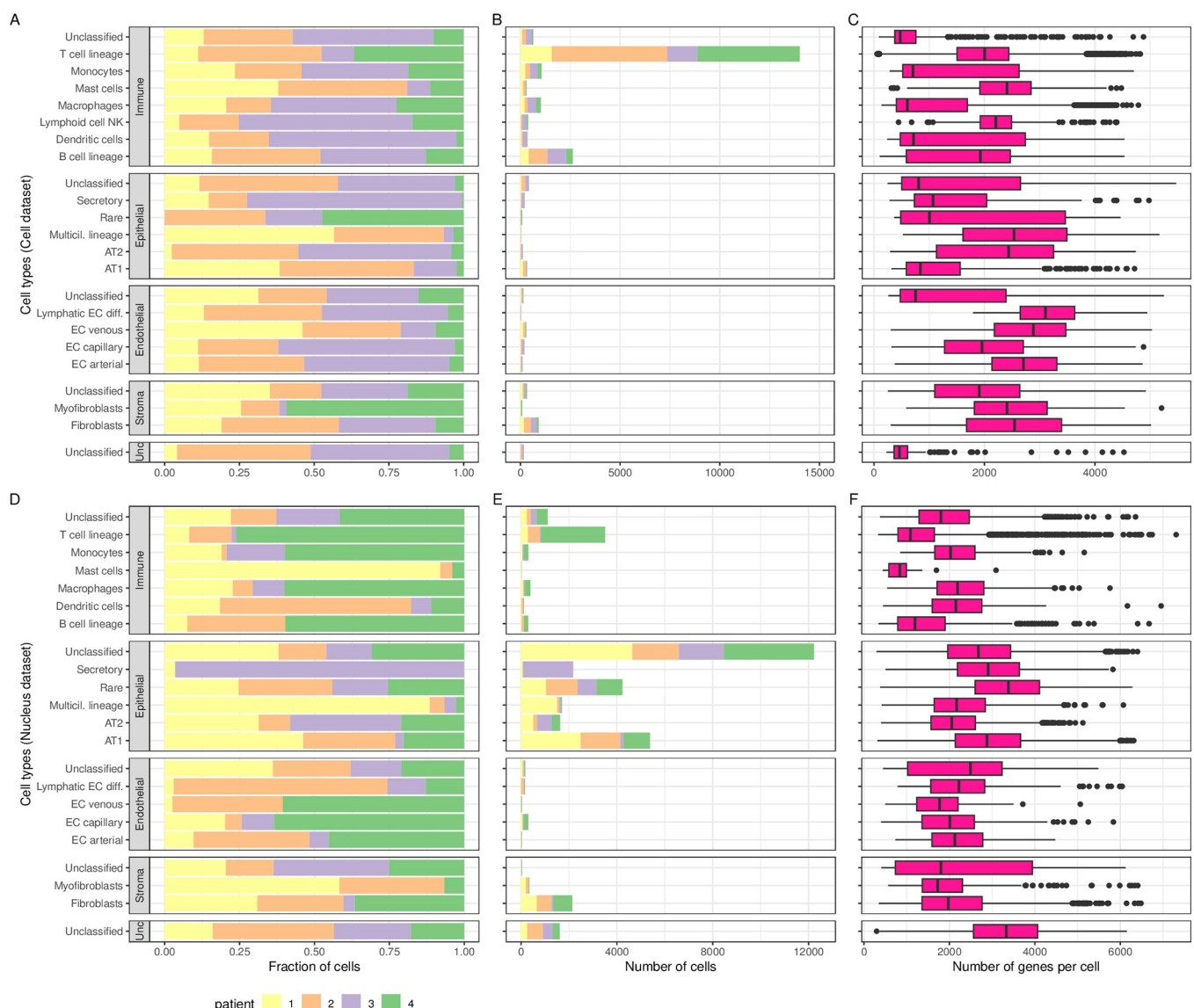

**Fig 6. Cell types characteristics of Tumor tissue.** For each of the four coarse (level 1) cell types annotations (*Immune*, *Epithelial*, *Endothelial*, *Stroma*) and unclassified (*unc*), further refined into finer categories (level 3 cell types), the fraction of cells (**A**: *Cell* samples, **D**: *Nucleus* samples) and the number of cells (**B**: *Cell*, **E**: N*ucleus*) originating from each patient. Box plots of the number of genes expressed (**C**: *Cell*, **F**: *Nucleus*), with plot center, box and whiskers corresponding to median, IQR and 1.5 × IQR, respectively. Note that only cell types with > 20 cells were retained for clarity in this visual representation.

stroma cells and endothelial cell types were relatively rare. With respect to epithelial cells, these were mainly composed of unclassified and AT1 in both *Cell* and *Nucleus* datasets, and secretory epithelial cells appeared to be mainly segregated to patient 3. However, rare cell types were much more common in the *Nucleus* than the *Cell* datasets.

## The cellular transition to LUAD

Given the known epithelial origin of lung adenocarcinoma and the role of the immune system in controlling the growth of carcinoma cells, we analysed the transition in the proportions of epithelial and immune cells from normal to adenocarcinoma tissue (**Fig 7A and 7B**). AT1, AT2 and multiciliated cells decreased in relative abundance in adenocarcinomas, and this was

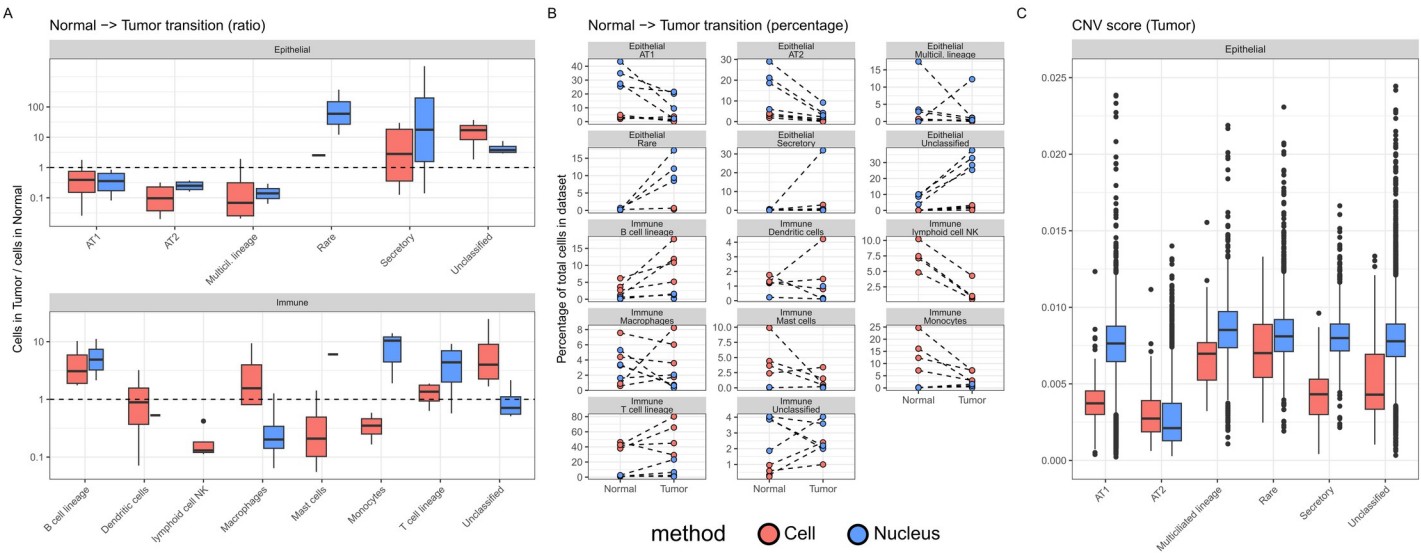

**Fig 7. Normal—tumor transition. A**: For each specific (level 3) Epithelial or Immune cell type, the fraction of cells they represent in the Tumor dataset divided by the fraction of cells they represent in the Normal dataset (ratios above 1 represent an increase in the Tumor dataset), with plot center, box and whiskers corresponding to median, IQR and $1.5 \times$ IQR, respectively **B**: The percentage of specific (level 3) Epithelial or Immune cell types in Tumor and Normal dataset. Each dot represents a patient and the dashed lines show the transition from Normal to Tumor for each patient. Note that only cell types with > 20 cells were retained for clarity in this visual representation. **C**: Box plots of the CNV score, with plot center, box and whiskers corresponding to median, IQR and $1.5 \times$ IQR, respectively.

consistent for the *Cell* and *Nucleus* datasets. On the contrary, rare, secretory and unclassified epithelial cell types increased in abundance in adenocarcinoma tissue in a consistent manner between *Cell* and *Nucleus* datasets. For Immune cells, patterns were harder to interpret given the small number of immune cells in the *Nucleus* dataset. Nevertheless, an augmentation of B and T cell lineages in adenocarcinoma was typically found for both datasets, as well as a drop in natural killer cells in the *Cell* dataset, while a discordant pattern was observed in monocytes. For macrophages, no consistent pattern was found in the transition from Normal to Tumor. When analysing more specifically interstitial macrophages (level 4 annotation), we confirmed a consistent augmentation in Tumor samples in *Cell* and *Nucleus* that was corroborated by immunohistochemical staining (**S4C Fig**).

We defined a genome-wide summary CNV score that relies on gene expression levels to identify gene deletion and duplication and aneuploid epithelial cells [26]. This score was the highest for multiciliated lineage and rare epithelial cell types, and the lowest for AT2 cells in the *Cell* and *Nucleus* dataset (**Fig 7C**). In addition, we also noted that annotation scores were negatively correlated with CNV scores for *Cell* ($r^2 = 0.11$, *p*-value < 2e-16) and *Nucleus* ($r^2 = 0.05$, *p*-value < 2e-16) datasets (**S7 Fig**). Finally, the inferred malignant classification of cells based on high CNV score and low annotation score demonstrated that the proportion of cancer cells in epithelial lineages was patient-specific and not always consistent between *Cell* and *Nucleus* (**S8 Fig**).

## Gene expression analyses

Using a pseudobulk method, we showed that aggregated gene expression correlates well between methods within tissues (r = 0.84 and 0.86) and between tissues within methods (r = 0.90 and 0.95, **Fig 8A**). Then, we showed in a dendrogram based on nuclear and whole-cell transcriptome data that samples cluster first by method (**Fig 8B**). The difference (DEGs)

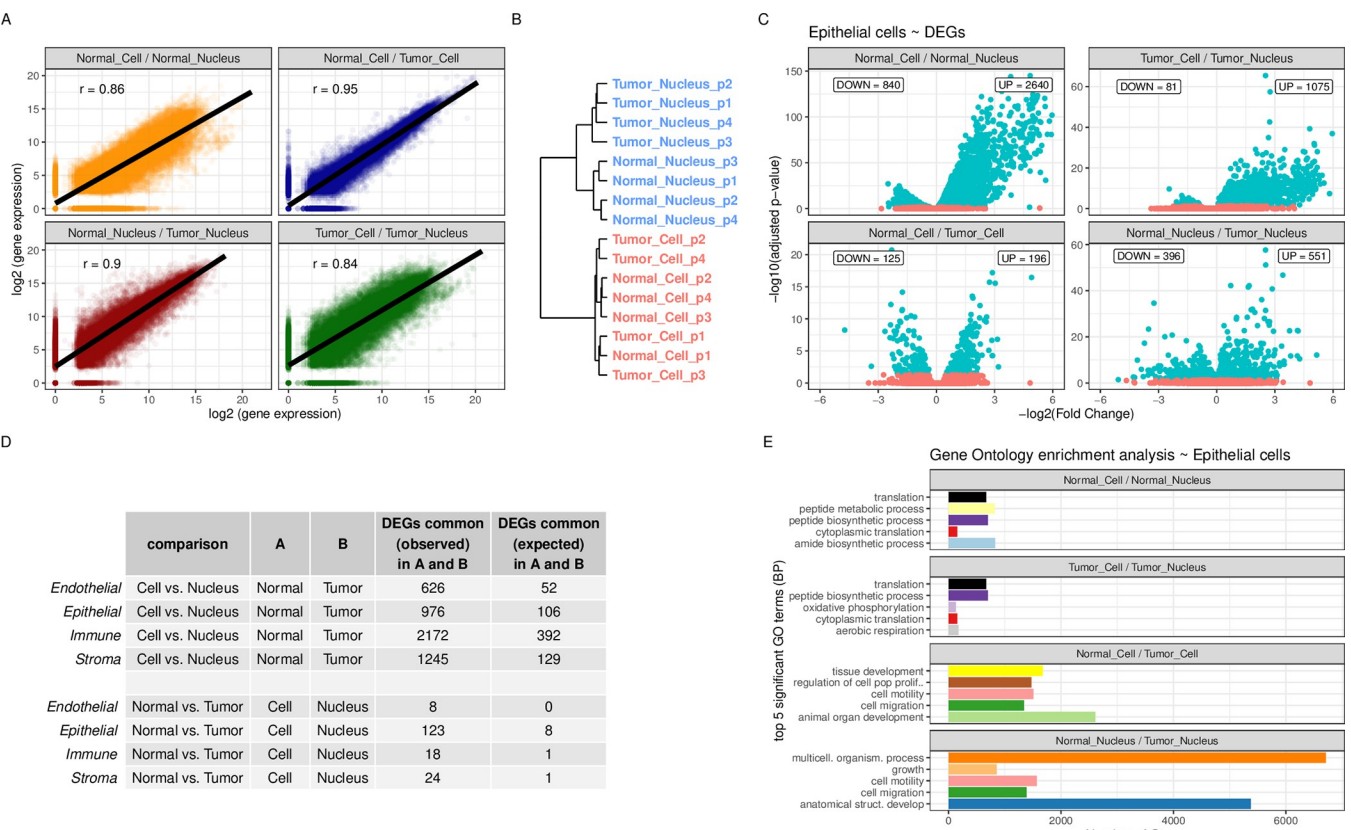

**Fig 8. Gene expression analyses per cell type. A**: Correlation in pseudobulk (aggregated) gene expression among datasets. On X-axis is log2 (gene expression) of first term in title (e.g. Normal *Cell* samples) compared to second term (e.g. Normal *Nucleus* samples) on y-axis **B**: Hierarchical clustering of top 5% most variable genes for *Cell* and *Nucleus* samples. **C**: significant DEGs (adjusted p-value < 0.05) for epithelial cells (in turquoise) in the four comparisons with the number of up-regulated and down-regulated genes in the first term in the title (e.g. Normal *Cell*). **D**: DEGs in common for *Cell* vs *Nucleus* in Normal (A) and Tumor (B) and for Normal vs. Tumor in *Cell* (A) vs *Nucleus* (B) **E**: Top five most significantly enriched gene ontology terms (Biological Process).

for epithelial cells between *Cell* vs. *Nucleus* in either Normal or Tumor (3,480 and 1,156 DEGs respectively, **Fig 8C**) was greater than between Normal vs. Tumor using the same method (321 and 947 DEGs respectively, **Fig 8C**). For both comparisons (*Cell* vs. *Nucleus* & Normal vs. Tumor), there were more DEGs in common across methods and tissues than expected by chance (**Fig 8D**, see **Tables D-G—in S1 Text** for full list of DEGs). In addition, looking at the five most significant enriched Gene Ontology, we saw that between *Cell* and *Nucleus*, similar GO terms were found (**Fig 8E**). These Biological Processes were related to mRNA translation, peptide biosynthesis and mitochondrial (aerobic) respiration. GO terms for the comparison Normal vs. Tumor were also partly concordant between *Cell* and *Nucleus* and all related to growth, development and migration (see **Table H in S1 Text** for other GO terms). DEGs for endothelial, immune and stromal cells are illustrated in **S9 Fig**.

Then using a Principal Component Analysis on the 39 markers genes commonly used to distinguish between Immune, Epithelial, Endothelial and Stromal cell types (see **S2 Fig** and Sikkema et al.[4]), we showed that these canonical markers genes match well with the reference-based annotation of the samples (**S10A Fig**). This confirms the validity of the reference-based method we used to annotate our samples. In addition, we showed no bias in the clustering of the samples based on the patient identity (**S10B Fig**). Instead, as we showed in **S10B Fig**, samples cluster according to the method (*Cell* vs. *Nucleus*) first, and more subtly based on the

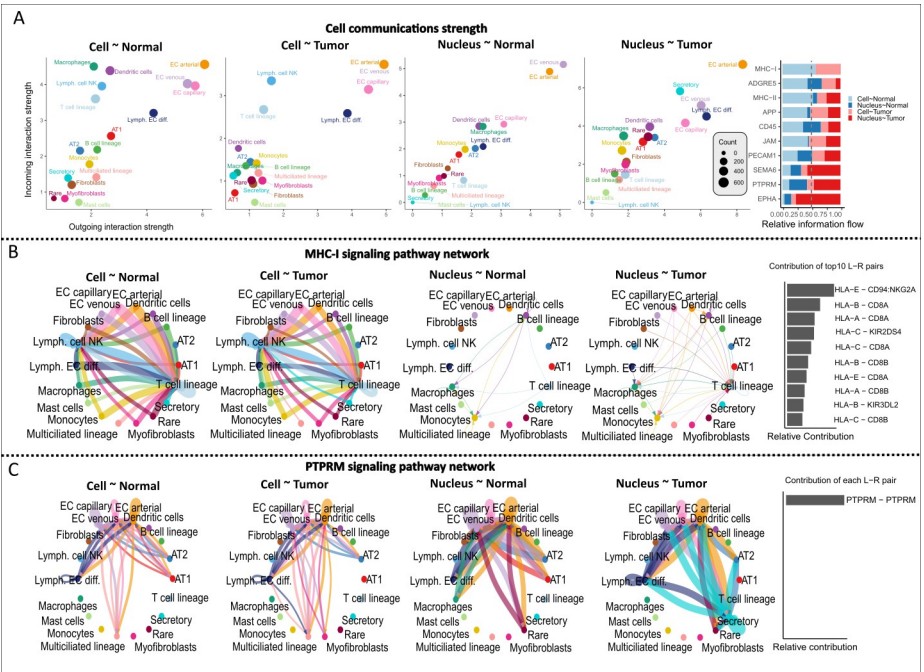

**Fig 9. The ligand-receptor interactome. A**: Scatter plots of ingoing and outgoing interactions per tissue type and method for common cell types (see methods) among all comparisons. To the right are the top 10 interacting pathways. **B**: An example of pathway common in *Cell*, rare in *Nucleus* (MHC-I) with the contribution of the top10 ligand-receptor interacting genes (bar plot to the right). **C**: An example of pathway rare in *Cell*, common in *Nucleus* (PTPRM) with the ligand-receptor interacting gene (bar plot to the right).

tissue effect (Normal vs. Tumor, **S10C Fig**). Based on Principal Components 3 and 4, we can see that for *Nucleus* samples, there is a better separation of Normal and Tumor samples, compared to the *Cell* samples **S10D Fig**), at least based on these 39 cell type markers genes. Finally, much like in the reference-based approach (**Fig 2**), the markers genes were less efficient in distinguishing between cell types in the *Nucleus* samples (**S10C Fig**).

## The ligand-receptor interactome differs between scRNA and snRNA

In **Fig 9A**, we visualised the incoming and outcoming interactions among 319 ligand-receptor interactions (cell-cell contact) for the *Cell*-Normal dataset. The number of interactions between cell types varies first according to the *Cell* vs. *Nucleus* methods (two-way ANOVA, $F = 90.7$, $p$-value $< 2e\text{-}16$) and then the Normal vs. Tumor tissue types ($F = 68.2$, $p$-value $= 3.6e\text{-}16$). In **Fig 9B**, we show an example of a typical pathway common in *Cell*, rare in *Nucleus* (Major Histocompatibility Complex-I) and its interacting genes, which is more similar between Normal vs Tumor tissue of the same experimental method (*Cell* vs *Nucleus*). An example pathway, rare in *Cell* but common in *Nucleus* (Protein Tyrosine Phosphatase Receptor Type M) and its self interacting gene is presented in **Fig 9C**. In this case, each network shows differences according to both the experimental method and tissue.

## The effect of immune depletion on Cell sequencing

In order to diminish the impact of the enrichment in immune cells induced by the single-cell dissociation protocol, we performed immune depletion in Normal and Tumor single-cell suspensions. We confirmed that the *Immune-depleted cell* dataset was enriched in epithelial cells

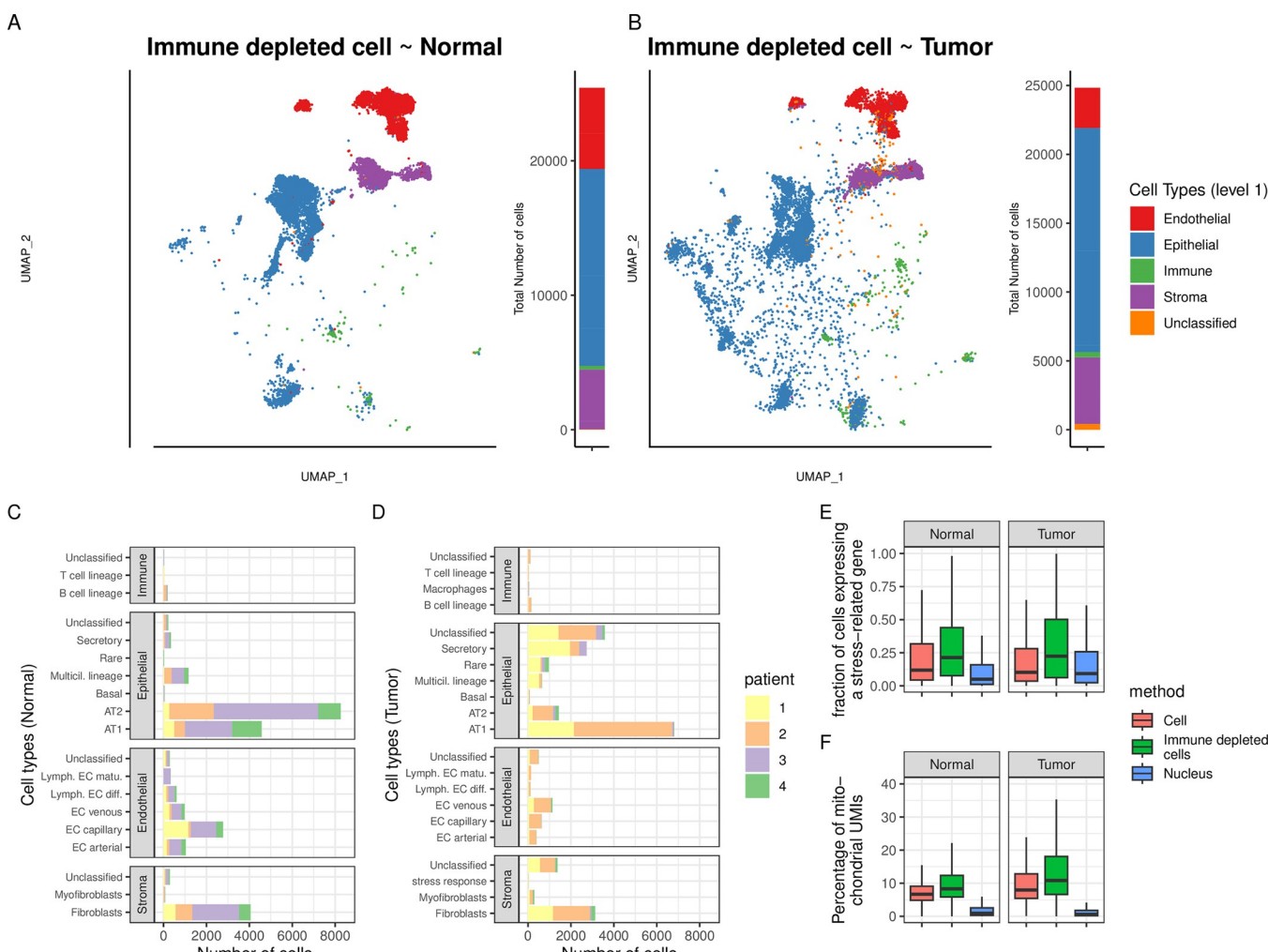

**Fig 10. UMAP representations and cell types annotations (*Immune-depleted cell*)** for Normal (**A**) and Tumor (**B**) tissue samples with general cell types (level 1) annotation. To the right of each UMAP, stacked bar plots indicate the proportion of each cell type in the specific dataset. Number of cells in the Normal (**C**) and Tumor (**D**) tissues, per patient. **E**: The percentage of cells expressing a stress-related gene signature as a function of the experimental method and tissue type. **F**: Percentage of sequencing reads (UMIs) assigned to mitochondrial genes as a function of tissue type and experimental method for unfiltered raw data.

and depleted in immune cells (**Fig 10A and 10B**). As such, both the Normal and Tumor tissues resemble the *Nucleus* dataset in the fact that they harbor a majority of epithelial cells (61.5% and 69.9% of total for the *Immune-depleted cell* and *Nucleus* dataset, respectively), yet they differ given that *Immune-depleted cell* harbors proportionally more endothelial (17.8% vs 4%) and stromal (18.4% vs 7.9%) cell types, but less immune cells (1.3% vs 13.0%). In addition, Normal tissues were largely composed of epithelial AT1 and AT2, while Tumor tissues also harbored secretory, rare and unclassified cell types, much like the *Nucleus* dataset (**Fig 10C and 10D**). Finally, as we observed for the non-depleted dataset, we saw an increase in the heterogeneity from Normal to Tumor datasets (median Silhouette index for each level 3 cell type annotation: $s_{i\ (Normal)} = 0.56$, median $s_{i\ (Tumor)} = 0.2$, two-way ANOVA, *p*-value < 2e-16, **S6 Fig**).

Next, we conducted Principal Component Analyses for each cell type on a representative subsample (top 5% most variables) of genes (Normal tissue). Based on this, *Immune-depleted-*

*cell* samples showed more variation between patients than *Cell* or *Nucleus* samples (**S11A–S11D Fig**). In addition, especially for immune cells, their overall gene expression signal differed from *Cell* and *Nucleus* samples (**S11A Fig**). Consequently, this implies that the remaining fraction of immune cells in *Immune-depleted cell* samples resemble the *Nucleus* samples.

Finally, we downloaded a set of 512 heat shock and stress response genes that were previously identified as affected by the scRNA-seq method [9]. Ninety four percent (482 genes) of the genes in this core dataset were also present in our current dataset, with varying levels of expression. More specifically, the percentage of cells expressing these genes was dependent on the method (**Fig 10E**, two-way ANOVA, *p*-value < 2e-16). The *Immune-depleted cell* dataset showed the highest expression of the stress response genes, whereas on average a cell from the *Immune-depleted cell* dataset expressed 21% of the 482 genes, compared to 11.0% and 6.9% for the *Cell* and *Nucleus* dataset, respectively. In addition, the proportions of cells expressing this core set of stress response genes were slightly, but significantly (*p*-value = 9.7e-8) higher in Tumor than in Normal tissues (12.4% and 11.5%, respectively). In a similar manner, higher mitochondrial contamination is often considered a sign of lower cell quality or viability [27] and we observed that the percentage of unique sequences (UMIs) assigned to mitochondrial genes in the raw data prior to any filtering was significantly higher (two-way ANOVA, *p*-value = 3.6e-5) in the *Immune-depleted cell* (mean = 15.2%) and *Cell* (11.2%) compared to the *Nucleus* (2.6%) dataset, while the tissue type (*p*-value = 0.10) had no significant effect (**Fig 10F**).

## Discussion

In this study we generated a dataset of 160,621 cells/nuclei showing commonalities and discordances in biological insights derived from single-cell and single-nucleus RNA-sequencing of paired normal-adenocarcinoma human lung specimens. A distinct portrait of cellular composition was observed per experimental methods that favors scRNA-seq of fresh samples to map the immune landscape of lung adenocarcinoma. On the other hand, snRNA-seq of frozen samples surpassed the relative merits of scRNA-seq to obtain a dataset with cell type proportion that match tissue content and to provide a more cost-effective approach for research applications necessitating a higher number of epithelial and cancer cells (**see Table I in S1 Text** for a summary of the benefits of each method). In these paired lung samples, we identified gene expression and cell type transitions from normal to tumoral tissue that were not always concordant whether cells or nuclei were examined. The most striking difference was the ligand-receptor interactions that varied more across methods (cells vs. nuclei) rather than tissue types (Normal vs. Tumor). Immune cell depletion partly alleviated some of the differences in cell type composition between cells and nuclei, but at the detriment of inducing a stress response and affecting the transcriptome biological signal. Finally, our analysis revealed that the recently proposed five-level hierarchical cell type annotation system by the Human Lung Cell Atlas [4] will require customization for assigning cell types specifically for tumor and nuclei samples.

Despite the fact that samples originated from the same patients' specimens, scRNA-seq and snRNA-seq varied substantially in their recovered cellular compositions and transcriptional landscape, thus highlighting the considerable impact of methodology on biological inference. While it has been shown previously that cryopreservation of tissue sample (such as performed for snRNA-seq) results in a major loss of epithelial cell types and an underrepresentation of T, B, and NK lymphocytes in the single-nucleus libraries [12,14], it is not necessarily apparent which experimental method is more biologically relevant. Slyper et al. [12] have suggested to analyse both fresh and frozen tissue, but this is often unrealistic in practice. For their part,

Denisenko et al. [14] indicated that the apparent discordance in the recovered cellular composition between scRNA and snRNA might be due to either an under-representation of immune cells in snRNA, or an under-representation of other cell types in scRNA due to incomplete dissociation. Andrews *et al.* [17] compared cells and nuclei of matched healthy human liver and concluded that cell-type frequencies were distorted in scRNA-seq. Early pioneering work in lung histology would suggest the same, whereas cell staining and electron microscopy has revealed that the alveolar regions of normal human lungs are comprised mainly of epithelial, endothelial and interstitial cells, while immune cells (macrophages) comprised a small fraction (~5%) of all cells identified [28]. We corroborated this observation with IHC staining in our matched Normal and LUAD samples. We thus conclude that in the context of lung adenocarcinoma and patient-matched normal samples, snRNA-seq provides a dataset comprising cell populations more closely matching tissue content.

We observed a decrease in cell viability in both depleted and non-depleted scRNA-seq, likely due to the longer sample preparation times at room temperature. While this could be partly alleviated by cold-activated proteases [9], it favors snRNA-seq as an experimental protocol to preserve sample integrity. Although immune depletion works well for removing immune cells and therefore might draw a more accurate representation of the lung cellular composition that is closer to snRNA-seq, it requires extra laboratory manipulations and has the adverse effect of affecting both cell viability (**Fig 10F**) and inducing a dissociation transcriptional stress response (**Fig 10E**), as shown previously [13].

The reference-based annotation used here provides an attractive alternative to unsupervised analysis [29]. We annotated the large majority of cells/nuclei in all tissue types, methods and patients (**Figs 2 and S1**) while showing that it performed as well as a marker-based approach, at least at the coarsest annotation level (**S2 and S1A Figs**). In their recent work comparing patient-matched lung adenocarcinoma samples, Trinks and colleagues used a similar statistical approach to annotate their snRNA-seq samples [30]. Arguably, the confidence in this reference-based annotation approach depends on several factors. Notably, the comprehensiveness of the reference, the quality and type of query data and the level of cellular granularity required to answer the biological question of interest will dictate the best approach to use. Nevertheless, an unsupervised-marker based approach also depends on several factors such as the clustering algorithm, the gene markers used, and almost always, the expertise and subjectivity of the person annotating the dataset [31,32]. Here, annotation and mapping were done using the same analytical framework for all samples and therefore provided an objective overview of the transcriptional cellular landscape. Fortunately, we were able to use a recently published comprehensive atlas of the lung (HLCA) [4], although such thorough cell atlases might not exist for all tissue types, biological conditions and demographic states [33]. The lower annotation scores observed in Nucleus and Tumor samples and consequently the greater number of unclassified cells, especially at the finer annotation levels suggest that these cells or nuclei have a distinct signature from the current reference cell type, much like we saw when conducting Principal Component Analysis of gene expression markers. A comparable phenomenon was also observed in the HLCA for different disease states [4] and the authors concluded that the HLCA must be viewed as a live resource that will require continuous updates in the future, including samples of diverse ethnic, clinical and experimental (e.g. snRNA-seq) backgrounds.

During the transition from normal to tumoral tissue, we identified a drop in AT1, AT2 and NK cells, concurrently with a rise in immune B and T cells, as previously identified [2,6,18]. In addition, tumoral cells showed an increased transcriptomic heterogeneity and a greater prevalence of copy number variants in epithelial cells. Similarly, it has been described that NSCLC exhibit important interpatient histologic heterogeneity and inferred origin of tumor cells [34]. Here, we showed that epithelial multiciliated lineages and rare cell types had higher Copy

Number Variants scores than other epithelial cell types, and the classification of cell malignancy confirmed patient-specific perturbations as previously reported [22]. Yet, the distinction between these epithelial cells is not always straightforward, especially in a context of oncogenesis. Along those lines, we noted that annotation scores were negatively correlated with CNV scores which implies that cells with high CNV (likely carcinoma cells) loose their cellular identity and become harder to classify as distinct lung cell types. During the construction of the HLCA, Sikkema *et al.*[4] also noted that a significant fraction of cells from adenocarcinomas did not cluster into the specific fine level cell types. Similarly, Wang *et al.*[24] argued that cancer cells originate from 'AT2-like' cells, but also nuanced this fact and stated that these form a distinct cluster from regular AT2 cells and have a transcriptional profile closely resembling other epithelial cells. Again, a more refined and thorough reference database will help to solve these questions.

Using a pseudobulk method, we showed better correlation of gene expression between cells and nuclei than previously reported RNA sequencing comparing isolated cells and nuclei (r between 0.53–0.74 by Barthelson and colleagues [35]), potentially because of our matched experimental design and improvements in single cell/nucleus sequencing in recent years. While we saw a large number of DEGs between cells and nuclei, there is also a lot of concordance in the DEGs identified in Normal and Tumor tissues. Previous studies reported that genes related to essential cell processes, taking place outside of the nucleus, such as ribosome- and mitochondrial-related genes, differ in expression between Single-Cell and Single-Nucleus sequencing [16,35]. Interestingly, there is also concordance in GO terms when comparing Normal and Tumor samples in *Cell* or *Nucleus* sequencing, but these processes have to do more with cell motility, migration and development.

This study has methodological implications as the literature and data comparing scRNA and snRNA are still scarce. Previous studies have compared scRNA and snRNA methods, but data from the same specimens were not necessarily available [11–13]. Head-to-head comparisons with the same specimens were performed using different platforms in mouse brain [15,16] and with 10x Genomics in mouse kidney [14]. In humans, we are only aware of one 10x study comparing matched scRNA and snRNA from human liver [17]. In the current study, we have both single-cell and single-nucleus on both normal lung and adenocarcinoma samples for all four patients and on the same platform (10x Genomics). Samples were resected in the same hospital and sequenced by the same laboratory. We thus have a unique and much-needed dataset to study the difference between single-cell and single-nucleus RNA-seq. By sharing our data with the scientific community, we aim to stimulate further comparisons between scRNA and snRNA, and allow others to build on our results.

Ultimately, we hope to develop a comprehensive transcriptional resource for the identification of cell-targeted biomarkers and therapeutic targets to treat and prevent LUAD and other ailing aspects of the lung. Accordingly, this study may have clinical significance as immunotherapy is currently revolutionizing the treatment of lung cancer. Response to immune checkpoint inhibitors relies on the existing cell-cell interactions between tumor and T cells (e.g., commercial immunotherapy drugs targeting the interaction between PD-1 in tumor cells and PD-L1 in T cells) [36] and identifying accurate biomarkers of response to immunotherapy is a major challenge in the field of lung cancer [37]. Consequently, this seems like a clinical problem where single-cell genomics can provide a solution. However, here we demonstrated that the ligand-receptor interactome landscape of lung adenocarcinoma is largely different whether cells or nuclei are evaluated. This may lead to conflicting prediction response to these novel immunotherapy agents. Accordingly, at least in the context of lung cancer, the choice between scRNA-seq and snRNA-seq has important implications. Our results favor scRNA-seq on fresh samples to provide a more comprehensive portray and granularity of the immune cells

diversity. This is consistent with the recommendation of using scRNA-seq to investigate immune populations in the human liver [17]. On the other hand, scRNA-seq may not be representative of the true cellular community, and lead to fewer difficult-to-dissociate tumor cells to assess relevant tumor-immune interactions. More studies will be needed to assess the best methods as well as to overcome other barriers to move single-cell genomics into the clinical setting [38].

## Materials and methods

### Ethics statement

All patients provided written informed consent, and the ethics committee of the IUCPQ-UL approved the study.

### Patients and samples

Lung samples were collected from four patients that underwent curative intent primary lung cancer surgery at the *Institut universitaire de cardiologie et de pneumologie de Québec–Université Laval* (IUCPQ-UL) in 2021–2023, henceforth referred to patient 1, 2, 3 and 4. The four patients were self-reported white French Canadian (European ancestry) with no prior chemotherapy and/or radiation therapy, and all patients were between the age of 59 and 69, former smokers with adenocarcinomas (See **Fig 1** for overview of experimental design, and **Table A in S1 Text** for detailed clinical characteristics of patients).

Following surgery, the explanted lobes were immediately transferred to the pathology department. For each patient, two ~1 cm$^3$ fresh Tumor samples and two ~1 cm$^3$ non-tumor (Normal) lung samples located distant from the tumor were harvested. The first set of tumor/non-tumor samples was transferred in dedicated tubes containing ice-cold RPMI (ThermoFisher, Cat. 11875093) for immediate cell dissociation and single-cell RNA sequencing (scRNA-seq) experiment. The second set of tumor/non-tumor samples was transferred in dedicated tubes, immediately snap-frozen in liquid nitrogen and stored at -80˚C until the day of the single-nucleus RNA sequencing (snRNA-seq) experiment. Lung tissue samples were obtained in accordance with the Institutional Review Board guidelines.

### Histologic evaluation

A thoracic pathologist (P.J.) reviewed each tumor and non-tumor hematoxylin and eosin (H&E) histology slides to confirm the presence/absence of tumor. Sections of 4.0 μm thick were cut from the selected blocks on a microtome and placed on charged slides. The following antibodies were used for IHC experiments: cytokeratin (monoclonal, clone AE1/AE3; Dako Agilent Technologies, Santa Clara, CA, USA), CD45 (monoclonal, clone DB11; Dako Agilent Technologies) and CD68 (monoclonal, clone PG-M1; Dako Agilent Technologies). All slides underwent heat-induced epitope retrieval in a Dako PT-Link using EnVision FLEX Target Retrieval Solution, high pH (9) Tris/EDTA buffer (Dako, Agilent Technologies), followed by an automatized IHC protocol on Dako Autostainer Link 48, using the EnVision FLEX+ kit reagents.

All H&E and IHC slides were digitized at 20X magnification with a slide scanner (NanoZoomer 2.0-HT; Hamamatsu, Bridgewater, NJ, USA). Slides visualization, cell segmentation and quantification were performed using QuPath (Version 0.5.1; The Queen's University of Belfast, Northern Ireland). Three different zones representing at least 50% of the whole surface area of the tissue were selected and analyzed. The numbers of positive cells were determined

using the automated cell detection tool and then visually validated by a pathologist (P.J.) for each marker.

## Sample preparation for scRNA-seq

Immediately after collection, the weight of each sample was recorded. Samples were transferred to 6-well cell culture plates, washed twice with 3 mL ice-cold PBS (Thermo Fisher, cat. 10010023) to remove excess blood and transferred to a 5 mL glass beaker. Using a 1 mL syringe and 25G needle, 300 μL of Enzyme dissociation mix was injected in the tissue followed by mechanical mincing into small fragments ($<1$ mm$^3$) using spring scissors for 2 minutes. Samples were then transferred to 50 mL Falcon tubes containing 5.7 mL of Enzyme dissociation mix and pipette mixed 5 times using wide bore 1 mL tips. The enzymatic digestion was performed at 37°C, using a Vari-Mix test tube rocker at max speed for 35 minutes. Samples were pipette mixed 20 times after 15 and 30 minutes using wide bore 1 mL tips. Enzyme dissociation mix contained: Pronase 1250 μg/mL (Sigma Aldrich, cat. 10165921001), Elastase 18.4 μg/ml (Worthington Biochemical, cat. LS006363), DNase I 100 μg/mL (Sigma Aldrich, cat. 11284932001), Dispase 100 μg/mL (Worthington Biochemical, cat. LS02100), Collagenase A 1500 μg/mL (Sigma Aldrich, cat.10103578001) and Collagenase IV 100 μg/mL (Worthington Biochemical, cat. LS 004186) in HBSS (Thermo Fisher, cat. 14170112). Enzymatic digestion was stopped by adding 1.5 mL of fetal bovine serum (FBS, ThermoFisher, cat. A3840301) followed by pipette mix 5 times using wide bore 1 mL tips. Dissociated cells were filtered through a 70 μm strainer and washed with 7.5 mL ice-cold PBS. Cells were then pelleted at 400g, 4°C for five minutes and supernatant was removed. Three cycles of red blood cells removal were performed as follow: cell pellet resuspended by manual agitation in 500 μL of ACK Lysis Buffer (ThermoFisher, cat. A1049201) and incubated on ice one minute. One mL of ice-cold PBS was added and cells were centrifuged at 400g, 4°C for two minutes and the supernatant was removed. The final pellet was resuspended in 500 μL ice-cold-PBS containing 0.04% Bovine Serum Albumin (BSA, Sigma Aldrich Cat. A7284) and 10% FBS. Cell suspensions were successively passed through 100 μm, 70 μm and 40 μm strainer using quick spin to reach 400g to filtrate each sample. Samples were transferred to 2.0 mL low binding tubes and kept at 4°C. Cell count and viability were performed using a 1:1 mix of cell suspension, Trypan blue (ThermoFisher, cat. 15250061), haemocytometer and conventional light microscopy. Cells suspensions meeting the following criteria were accepted for scRNA-seq library preparation: absence of aggregated cells, a viability $>80\%$, and a total cell count between 400 and 1200 cells/μL. $1 \times 10^5$ cells were transferred to a low binding 2 mL tube and kept at 4°C (non-depleted fraction). The remaining cells (from 2 to $5 \times 10^6$ cells) were submitted to CD45+ immune cell depletion protocol (single cells depleted fraction) as described below. The characteristics of the lung specimen and the single cell suspension for each sample are given in **Table B in S1 Text**.

## CD45+ immune cell depletion

Cells (from 2 to $5 \times 10^6$ cells) were centrifuged at 300g, 4°C, 10 minutes. The supernatant was removed and the cell pellet was resuspended in 80 μL MACS buffer (0.5% BSA, 2 mM EDTA pH 8.0 in PBS) previously degassed for 1 hour at room temperature. Twenty μL of CD45 microbeads (Miltenyi Cat. 130-045-801) were added and sample was incubated 15 minutes at 4°C followed by addition of 1 mL MACS buffer and centrifugation 300g, 10 minutes at room temperature. Supernatant was removed and pellet resuspended in 2-steps 100 μL + 400 μL MACS buffer. The total volume (500 μL) was applied to a LS Positive Selection Column (Miltenyi Cat. 130-042-401) previously rinsed with 3 mL MACS buffer and installed on a MidiMACS magnetic Separator with a collection tube. Column was rinsed with 3 X 3 mL MACS buffer

and all volumes (9.5 mL) were collected which contained the CD45-negative fraction. CD45-negative cells were centrifuged 300g, 10 minutes at room temperature followed by supernatant removal. Cells were washed twice with 1 mL PBS followed by centrifugation at 300g, 10 minutes after each wash. Cells were finally resuspended in 100 μL BSA 0.04%, 10% FBS in PBS and kept at 4°C. Cell count and viability were performed using a 1:1 mix of cell suspension, Trypan blue, haemocytometer and conventional light microscopy. Cells suspensions meeting the following criteria were accepted for scRNA-seq library preparation: absence of aggregated cells, a viability >80%, and a total cell count between 400 and 1200 cells/μL.

## Sample preparation for snRNA-seq

Nuclei suspension was prepared from ~30 mg snap frozen tissue using Chromium Nuclei Isolation Kit as per manufacturer's protocol (10x Genomics Cat. 1000494). Nuclei count and integrity were performed using a 1:1 mix of nuclei suspension and methylene blue 0.25% (Ricca Chemical, Cat. 48504), haemocytometer and conventional light microscopy. Nuclei suspensions meeting the following criteria were accepted for snRNA-seq library preparation: absence of aggregated nuclei, nuclei with circular shape and intact membrane (without blebbing) >80%, and a total nucleus count between 400 and 1200 nuclei/μL. Nuclei suspensions were kept at 4°C until proceeding with 10x Genomics snRNA-Seq library preparation protocol.

## 10x Genomics sn/scRNA-seq library preparation

For each sample, approximatively 15,000 nuclei or cells were loaded into each channel of a Chromium Next Gel Beads-in-emulsion (GEM) Chip G (10x Genomics Cat. 1000127) as per manufacturer's instruction for GEM generation and barcoding. Given the cell capture efficiency of around 65%, 10,000 cells per library were therefore expected. The Chip was run on the Chromium Controller, GEMs were aspirated and transferred to a strip tube for cDNA synthesis, cDNA amplification and library construction using Chromium Next GEM single-cell 3' Library Kit v3.1 (10x Genomics Cat. 1000128) and Single Index Kit T Set A (10x Genomics Cat. 2000240) as per manufacturer's instruction. The library average fragment size and quantification was performed using Agilent Bioanalyzer High Sensitivity DNA kit (Agilent Cat. 5067–4626) and a final concentration determination was performed using NEBNext Library Quant Kit for Illumina (New England Biolabs Cat. E7630) prior to library sequencing.

## Next generation sequencing

Libraries were individually diluted to 10 nM, pooled and sequenced on an Illumina NextSeq 2000 system following manufacturer's recommendations. Sequencing was realized on a P3 (100 cycles) cartridge, aiming for 200 to 500 million reads per library (sample). Run parameters for paired-end sequencing were as follow: read 1, 28 nucleotides; read 2, 91 nucleotides; index 1, 8 nucleotides; and index 2, 0 nucleotide.

## Single cell/nucleus data preparation

Demultiplexing, alignment and transcript counting was performed using the *Cellranger* software (v7.1.0, 10x Genomics) on our local server (Lenovo ThinkSystem SR650, 40 cores and 384GB RAM). The BCL files from the Illumina sequencing run were first demultiplexed into FASTQ files using the *cellranger mkfastq* command. Read alignment and UMI counting were then executed with the *cellranger count* command (see alignment and cell statistics in **Table J**

in S1 Text). We used GRCh38 as the reference transcriptome available on Gencode, release 43 (GRCh38.p13).

## Data quality control

The most up-to-date bioinformatics procedure defined by the R (v4.3.3) [39] library *Seurat* (v5.0.2) [27] was used to create an object for each sample and calculate values for *nCount* (number of Unique Molecular Identifiers [UMI] per cell), *nFeatures* (number of genes expressed per cell) and *percent.mt* (fraction of UMIs aligning to mitochondrial genes) parameters. Using the R library *scuttle* (v1.10.1) [40], we determined outlier values for *nCount*, *nFeatures* and *percent.mt* based on the median absolute deviation and sub-set each sample accordingly. Note that for the *percent.mt* parameter, if necessary, we further capped this outlier value at twenty-five percent per sample.

For each sample, we then performed normalization and variance stabilization using the function *SCTransform*, which also has the benefit to regress out the *percent.mt* effect from the underlying count data. Then, using the R library *DoubletFinder* (v2.0.3) [41], we identified and removed doublets (assuming a five percent doublet rate), which occur when multiple cells are captured into a single oil droplet during the GEM generation.

## Reference-based cell type annotation and mapping

On each of these curated samples, cellular annotation was performed using the R library *Azimuth* (v0.4.6) [29] and the most recent version of the Human Lung Cell Atlas (HLCA v2) [4]. Note that in the subsequent methodology, *cell* annotation refers to the annotation of a uniquely barcoded GEM sample stemming from either a scRNA-seq or a snRNA-seq dataset.

The HLCA is a comprehensive and curated reference dataset constructed using a diverse set of 107 healthy lung samples (584,444 cells) and which allows to identify the transcriptional signature of 61 hierarchical cell types, from the coarsest possible annotations (level 1: *Immune*, *Epithelial*, *Endothelial* and *Stroma*), recursively broken down into finer levels (levels 2–5). In addition, this reference-based mapping approach allows to robustly and sensitively compare samples of broad cellular compositions, while also identifying specific and rare cell populations [27,29,42].

Specifically, for each sample (query), the algorithmic approach first identifies anchors between the reference and query (that is, pairs of cells from each dataset that are contained within each other's neighborhoods) and uses these anchors to integrate the query dataset onto the reference. Then, the embeddings of the query data onto the reference Principal Components (50 PCs) are calculated and visualised directly onto the reference two-dimensional Uniform Manifold Approximation and Projection (UMAP). Finally, annotation scores [0:1], which reflect the confidence in the annotation, were used to label cell types, whereas cells with annotation scores < 0.5 were labelled as *unclassified*.

## Copy number variations analysis

For each patient, we performed an analysis of Copy-Number Variants (CNVs) in order to identify epithelial aneuploid cells based on the premise that gene CNVs can be identified using the difference between the mean log expression level of non-cancerous reference cells (here epithelial cells in the Normal tissue, either in *Cell* or *Nucleus* sequencing) and the log gene expression level of an epithelial cell of interest in the Tumor tissue. This was performed using the R library *infercnv* (v1.17.0) [26] and a general index (CNV score) for each cell was defined as the mean sum of square of scaled [-1;+1] standardized log fold-change values. Finally, we classified cells as malignant based on the integration of several parameters, as typically performed [22,25]. Cells of epithelial origin, with a high CNV score (top quintile), and a cell type

annotation score in the bottom quintile (malignant cells are typically more difficult to annotate due to the reprogramming of the lung adenocarcinoma transcriptome) were labelled as malignant. Consequently, this allowed an objective comparison of the malignant cells between methods and patients.

## Biological dataset comparisons

We integrated twenty-four samples into six different datasets (*Cell-Normal*, *Nucleus-Normal*, *Cell-Tumor*, *Nucleus-Tumor*, *Immune-depleted cell-Normal*, *Immune-depleted cell-Tumor*), in order to quantify biological similarities and differences among datasets (see **Fig 1D–1G** for summary of comparisons and accompanying figures). Given that the same reference dimensionality reduction (PCA) and visualisation space (UMAP) was used for each sample, we could simply merge expression data, metadata and projections into objects that account for technical variation among sample in order to quantify patterns. For each individual cell, we also calculated a Silhouette index [43] to evaluate the goodness of fit of the clustering, whereas the index is calculated from the UMAP embeddings and the clusters correspond to specific cell type (level 3) annotations. We then tested the effect of the experimental method and tissue type on the Silhouette index using a two-way Analysis of Variance (ANOVA).

## Gene expression analyses

Differentially expressed genes (DEGs) were identified using a pseudobulk approach, which has been shown to outperform other single-cell differential expression methods [44]. In this case, it first consists of aggregating (i.e. summing up) counts by cell type (epithelial, endothelial, immune and stroma) and quantifying the expression levels per gene but with respect to cell type, patient, tissue and method.

We then performed hierarchical clustering (Ward distance) on a subset of the top 5% most variable genes to illustrate the transcriptome wide effects of the methods and tissues. We quantified the total number of differently expressed genes (DEGs) per cell type, tissue and method using a negative binomial distribution (DESeq2 R Package, v 1.40.2) [45]. Specifically, we looked at the number of DEGs in common between methods of the same tissue and between tissues of the same method, to see how concordant they were compared to a null expectation (i.e. [number of DEGs in comparison A / number of genes in comparison A] X [number of DEGs in comparison B / number of genes in comparison B] X total number of genes). Finally, we performed enrichment analyses (Gene Ontology Biological Process) using the R package topGO [46] (v 2.52.0) to look at concordance in functional terms among DEGs.

In addition, we performed a principal component analyses (PCA) with the R library *FactoMineR* (v2.10) [47] of the normalized summed counts using the 39 markers genes typically used to distinguish the four major cell types (endothelial, epithelial, immune, stroma, see also **S2 Fig** for the list of markers genes from Sikkema *et al*. 2023 [4]). As such, each sample (four patients X two methods X two tissues) is represented by four data points based on its summed cell type specific component.

We also conducted PCA on the top 5% most variable genes in order to look at the clustering of *Cell*, *Nucleus* and *Immune-depleted cells* samples based on an overall gene expression signal for each coarse level 1 cell types.

## Ligand-receptor analysis

In order to infer and visualise the intercellular communication among cell populations, we used the R library *cellchat* (v 1.6.1) [48]. We quantified the cell-cell interaction pathways in Normal and Tumor tissues (*Cell* and *Nucleus* dataset) to describe the cellular transition during

oncogenesis and quantify how the experimental method and tissue type affected the results. We limited this analysis to level 3 annotation and excluded infrequent cell types (<500 cells in total) and cells that were unclassified at the level 3 annotation. We quantified the number of interactions from and to each cell type and tested the effect of the experimental method and tissue type using a two-way ANOVA.

### Stress-related genes

To quantify the effect of our *Cell*, *Nucleus* and *Immune-depleted cell* experimental methods on the overall stress responses of the cell populations, we analysed the expression pattern of a core set of 512 heat shock and stress response genes that were previously identified to be affected by the scRNA-seq sample preparation method [9]. We quantified the proportions of cells that expressed these genes for each sample and tested the effect of the experimental method, tissue type and patient using a two-way ANOVA.

### Supporting information

**S1 Fig. UMAP visualization of all 160,621 cells / nuclei** that passed quality control per level 3 annotation (**A**), tissue type (**B**), experimental method (**C**) and patient (**D**). (PNG)

**S2 Fig. UMAPs for the Cell (A) and Nucleus (F) dataset** with coarse level annotations and feature plots according to average expression level of the gene markers defined for each cell type by HLCA (see below), in *Cell* (**B-E**) and *Nucleus* (**G-J**). Immune-specific gene markers = 'LCP1','CD53','PTPRC','COTL1','CXCR4','GMFG','FCER1G','LAPTM5','SRGN','CD52' Epithelial-specific gene markers = 'KRT7','PIGR','ELF3','CYB5A','KRT8','KRT19','-TACSTD2','MUC1','S100A14','CXCL17' Endothelial-specific gene markers = 'PTRF','CLDN5','AQP1','PECAM1','NPDC1','VWF','GNG11','RAMP2','CLEC14A' Stromal-specific gene markers = 'TPM2','DCN','MGP','SPARC','CALD1','LUM','TAGL-N','IGFBP7','COL1A2','C1S' (TIF)

**S3 Fig. UMAP per patients for Normal samples.** (TIF)

**S4 Fig. A.** Hematoxylin and Eosin staining of Normal and Tumor lung parenchyma used for cell isolation. 100X magnification. **B.** Fraction of Epithelial (AE1/AE3) and Immune (CD45) cells identified through immunohistochemical staining compared to Epithelial and Immune cells (level 1), obtained for the three experimental methods, i.e. *Cell*, *Nucleus* and *Immune-depleted cell*. **C.** Number of macrophages (CD68) identified through immunohistochemical staining compared to the most relevant cell type (Interstitial macrophage, level 4) for the *Cell* and *Nucleus* datasets. The *Immune-depleted cell* dataset was excluded because the number of macrophages was insufficient. (TIF)

**S5 Fig. UMAP per patients for Tumor samples.** (TIF)

**S6 Fig. Silhouette index to evaluate the goodness of fit of the clustering.** For each cell / nucleus, Silhouette Indices are calculated from the UMAP embeddings and the clusters correspond to a specific cell type (level 3) annotations. Silhouette Index was significantly lower (less structured clusters) for *Tumor* rather than *Normal* samples. (TIF)

**S7 Fig. Annotation score (level 3) is negatively correlated with CNV score.** Data points were binned (50 hexagonal bins in x-axis * 50 hexagonal bins in y-axis) to reduce overplotting. (TIF)

**S8 Fig. The percentage of epithelial cells classified as malignant for each patient in *Cell* and *Nucleus* samples.** (TIF)

**S9 Fig. DEGs (in turquoise) for Endothelial, Immune and Stromal cells with the number of up-regulated and down-regulated genes.** (TIF)

**S10 Fig. Principal Component Analysis on the 39 markers genes used to distinguish between Immune, Epithelial, Endothelial and Stromal cell types (see S2 Fig legend for a list of markers genes used). A**. Marker genes loadings on the PCA (arrows colored by the cell type they are used to define) match well with the reference-based annotation of the samples (colored points). **B.** No bias in the clustering of the samples based on the patient identity. **C.** Samples cluster according to the method. Nucleus samples are closer to the center of the PCA, which implies that markers genes were less efficient in distinguishing between cell types in these samples. **D.** In Principal Components 3 and 4, Nucleus samples are separated by tissue type (Normal and Tumor). (TIF)

**S11 Fig.  Principal Component Analysis** on the top 5% most variable genes (Normal tissue) for **A.** Immune cells **B.** Epithelial cells **C.** Endothelial cells and **D.** Stromal cells. 95% confidence interval ellipses are drawn for each method based on all four patients. (TIF)

**S1 Text. Supporting information tables. Table A in S1 Text.** Demographic and clinical characteristics of the four patients analysed. Continuous variables are presented as mean ± SD. Discrete variables are presented as n (%). **Table B in S1 Text.** Characteristics of the lung specimens and single cell/nucleus suspensions. **Table C in S1 Text.** Number of cells/nuclei identified at each hierarchical level (level 1–5, 61 cell types defined at the finest level by the HLCA). Thirty-five finest level cell types were recovered with >100 cells (51 finest level cell types with at least one cell identified). Here unclassified refers to cells/nuclei which could not be assigned confidently to the specific annotation level (annotation score < 0.5). **Table D in S1 Text.** Differentially Expressed Genes (Normal Cell versus Normal Nucleus samples). **Table E in S1 Text.** Differentially Expressed Genes (Normal Cell versus Tumor Cell samples). **Table F in S1 Text.** Differentially Expressed Genes (Normal Nucleus versus Tumor Nucleus samples). **Table G in S1 Text.** Differentially Expressed Genes (Tumor Cell versus Tumor Nucleus samples). **Table H in S1 Text.** Differentially Expressed Gene Ontology terms (Biological Process). **Table I in S1 Text.** Benchmarking scRNA-seq and snRNA-seq methods in paired normal-adenocarcinoma lung samples using the 10x Genomics workflows. **Table J in S1 Text.** 10X Genomics Cell Ranger software—QC metrics. (XLSX)

## Acknowledgments

The authors would like to thank the research staff at the IUCPQ biobank for their valuable assistance.

## Author Contributions

**Conceptualization:** Sébastien Renaut, Patrice Desmeules, Sébastien Thériault, Patrick Mathieu, Philippe Joubert, Yohan Bossé.

**Data curation:** Sébastien Renaut, Victoria Saavedra Armero, Dominique K. Boudreau, Nathalie Gaudreault.

**Formal analysis:** Sébastien Renaut, Victoria Saavedra Armero, Dominique K. Boudreau, Nathalie Gaudreault, Philippe Joubert.

**Funding acquisition:** Patrice Desmeules, Sébastien Thériault, Patrick Mathieu, Philippe Joubert, Yohan Bossé.

**Investigation:** Sébastien Renaut, Yohan Bossé.

**Methodology:** Sébastien Renaut, Victoria Saavedra Armero, Dominique K. Boudreau, Nathalie Gaudreault, Yohan Bossé.

**Project administration:** Yohan Bossé.

**Software:** Dominique K. Boudreau.

**Supervision:** Yohan Bossé.

**Validation:** Yohan Bossé.

**Visualization:** Sébastien Renaut, Philippe Joubert.

**Writing – original draft:** Sébastien Renaut, Yohan Bossé.

**Writing – review & editing:** Sébastien Renaut, Victoria Saavedra Armero, Dominique K. Boudreau, Nathalie Gaudreault, Patrice Desmeules, Sébastien Thériault, Patrick Mathieu, Philippe Joubert, Yohan Bossé.

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
