## [Decision Letter · Decision Letter 0]

29 Feb 2024

Dear Dr. Bossé,

Thank you very much for submitting your Research Article entitled 'Single-cell and single-nucleus RNA-sequencing from paired normal-adenocarcinoma lung samples provides both common and discordant biological insights' to PLOS Genetics.

The manuscript was fully evaluated at the editorial level and by independent peer reviewers. The reviewers appreciated the attention to an important problem, but raised some substantial concerns about the current manuscript. Based on the reviews, we will not be able to accept this version of the manuscript, but we would be willing to review a much-revised version. We cannot, of course, promise publication at that time.

If you decide to revise the manuscript for further consideration at PLOS Genetics, please aim to resubmit within the next 60 days, unless it will take extra time to address the concerns of the reviewers, in which case we would appreciate an expected resubmission date by email to plosgenetics@plos.org.

We are sorry that we cannot be more positive about your manuscript at this stage. Please do not hesitate to contact us if you have any concerns or questions.

Yours sincerely,

Yan Tang

Guest Editor

PLOS Genetics

David Kwiatkowski

Section Editor

PLOS Genetics

Reviewer's Responses to Questions

**Comments to the Authors:**

Reviewer #1: In this manuscript, the authors compared single-cell RNA sequencing (scRNA-seq) and single-nuclei RNA sequencing (snRNA-seq) of paired normal-adenocarcinoma human lung samples to explore the similarities and differences in cell types, immune landscapes, and other aspects. The experimental design is appropriate and the technical content is high, providing valuable practical and academic insights for the field of medicine. However, the limitations of this paper lie in its purely descriptive nature and the lack of comprehensive details in result interpretation. Here are a few suggestions for improving the manuscript.

Question 1: In main text, article only provides explanations for Figures 1A and 1B. Please provide descriptions for all the main figures would enhance the clarity of the manuscript.

Question 2: Please standardize the sequential letter format in the main figure of the article.

Question 3: Line 363, please ensure consistent spacing between numbers and units throughout the document.

Question 4: Whether an attempt to perform integrated analysis of Cell and Nucleus datasets could be performed, which may provide a more comprehensive understanding of whether the differences between scRNA-seq and snRNA-seq are solely due to variations in cell proportions.

Question 5: Please analyze cellular communication and other relevant factors using Immune-depleted datasets to better elucidate how Immune-depleted cell datasets can mitigate the differences between Cell and Nucleus datasets, .

Question 6: The “Cell composition differs from Nucleus in Normal lung tissue” part mentions that the cellular landscapes differ greatly between Cell and Nucleus datasets, and proposes that in the context of lung adenocarcinoma and patient-matched normal samples, Nucleus provides a cell populations more closely matching tissue content. Whether HE staining or other experimental results could be provide to validate this conclusion?

Question 7: The “The cellular transition to lung adenocarcinoma” part mentions that there is inconsistency in the transition from normal to tumor in some immune cells between Cell and Nucleus datasets, Whether immunohistochemical staining or other experimental results could be provided to confirm this conclusion and compare the accuracy of the two sequencing methods?

Question 8: In “The effect of immune depletion on Cell sequencing” part, the impact of gene expression site on the results whether has been considered in the evaluation of stress-related genes?

Reviewer #2: Renaut et al. performed a head-to-head comparison between scRNA-seq and snRNA-seq in matched normal-adenocarcinoma human lung samples. They obtained more than 160,000 cells/nuclei from 4 patients and detected a significant difference in cell type proportions between scRNA-seq and snRNA-seq. They also revealed the cellular and molecular transition that occurs from normal lung tissue to adenocarcinoma, and evaluated the commonality and discordance in the biological insights gained from cells versus nuclei. Particularly, the ligand-receptor interactome landscape of lung adenocarcinoma varied significantly between scRNA-seq and snRNA-seq. Finally, they concluded that scRNA-seq is fit for the immune landscape mapping of lung adenocarcinoma, while snRNA-seq may capture more reasonable cell type proportion.

This is a comprehensive benchmarking study for the comparison between scRNA-seq and snRNA-seq in matched normal-adenocarcinoma human lung samples. The data provides a useful resource for the related studies, and the findings can help researches to choose appropriate method when dealing with similar issues. Several concerns remain to be addressed, as shown below.

Major:

1. In the introduction part, the authors should cite the previous study about liver scRNA-seq and snRNA-seq comparison, since this study also obtained similar conclusion (Single-Cell, Single-Nucleus, and Spatial RNA Sequencing of the Human Liver Identifies Cholangiocyte and Mesenchymal Heterogeneity, https://doi.org/10.1002/hep4.1854).

2. Considering the more dropout rate and less transcript abundance in snRNA-seq, it is maybe not appropriate to annotate the snRNA-seq dataset with the single-cell based reference, HLCA, though it is maybe fit for scRNA-seq dataset. As their results shown that more unclassified identities are presented in the snRNA-seq dataset. Thus, unsupervised classification is also needed to provide more accurate cell type annotation, and a separate annotation of scRNA-seq and snRNA-seq datasets is also a good way to double check the accuracy of the cell identities.

3. The author should also check whether their findings are shared by all the four patients or biased by individual patient, such as the results in Figure 3 and 5.

4. In the CNV analysis, the authors stated that other cell types rather than AT2 exhibited higher CNV scores. What are the normal reference cell types for inferring the CNV scores, as the normal reference cell types can greatly affect the result. I do not think it is appropriate to use immune cells to perform the CNV analysis, because malignant cells mainly originate from epithelial cells, thus the author should use normal epithelial cells as reference, including each normal epithelial cell type.

5. The authors didn’t identify the malignant cells in tumor samples neither in ‘Cell dataset’ nor ‘Nuleus dataset’, which should be one necessary step for analyzing tumor samples.

6. For cellular transition analysis, the authors mentioned that ‘For immune cells, patterns were harder to interpret given the small number of immune cells in the Nucleus dataset’. As shown in Figure 6B and F, the number of immune cells in ‘Nucleus dataset’ is bigger than that in ‘Cell dataset’, therefore, the interpretation is not accurate.

7. The authors compared the cell composition and cell communication difference between ‘Nuleus dataset’ and ‘Cell dataset’. The authors also should identify the molecular differences between the two datasets for specific cell types, which might identify some cell-type shared or specific molecular differences, giving some clues for studying biological insights difference between the two datasets.

Minor:

1. In the first sentence of the Abstract section, snRNA-seq should be ‘single-nucleus’ not ‘single-nuclei’.

2. In the first sentence of the Introduction section, single-cell transcriptomics should not be short for ‘scRNA-seq’, single-cell RNA-sequencing is more appropriate. Likewise, the full name for snRNA-seq should also be mentioned when it appears for the first time (Page 1, Line 67).

Reviewer #3: This study compares single-cell RNA-sequencing (scRNA-seq) and single nuclei RNA-sequencing (snRNA-seq) in matched normal-adenocarcinoma human lung samples with a total of 160,621 cells/nuclei. They mainly find that while scRNA-seq captures more immune cells in non-tumor lung and snRNA-seq captures more epithelial cells, the two methods show discordant cell type transitions from normal to tumoral tissue.

Although this study conducted a relatively wide range of experiments, my main concerns with this manuscript are that the authors did not analyze and discuss the experimental results deeply enough. Moreover, this study did not further explore and elucidate the development of lung adenocarcinoma with significant phenotypic changes and key gene expression changes using single cell and single nucleus sequencing techniques. In fact, as early as 2020, there were several papers that conducted similar experiments and analyses on kidney (PMID: 30510133), PBMCs (PMID: 32341560) and even the same non-small cell lung cancer samples (PMID: 32405060) as this study. As a result, this study does not show enough innovation and powerful insights compared to previous research work.

Main concerns:

1. Authors should use single cell and single nucleus sequencing techniques to further explore potential cell interaction changes or gene expression changes that may be associated with lung adenocarcinoma through a variety of downstream analytical tasks, such as gene differential expression or gene regulatory network analysis.

2. For the multiple experimental results of this manuscript, the author should carry out detailed analysis and discussion, rather than simply display them.

3. The impact of immune cell depletion on downstream analysis and interpretation is not fully characterized.

**Have all data underlying the figures and results presented in the manuscript been provided?**

Reviewer #1: None

Reviewer #2: Yes

Reviewer #3: Yes

PLOS authors have the option to publish the peer review history of their article (what does this mean?). If published, this will include your full peer review and any attached files.

Reviewer #1: No

Reviewer #2: **Yes: **Ji Dong

Reviewer #3: No

---

## [Decision Letter · Decision Letter 1]

13 May 2024

Dear Dr Renaut,

We are pleased to inform you that your manuscript entitled "Single-cell and single-nucleus RNA-sequencing from paired normal-adenocarcinoma lung samples provides both common and discordant biological insights" has been editorially accepted for publication in PLOS Genetics. Congratulations!

Yours sincerely,

Yan Tang

Guest Editor

PLOS Genetics

David Kwiatkowski

Section Editor

PLOS Genetics

Comments from the reviewers (if applicable):

Reviewer's Responses to Questions

**Comments to the Authors:**

Reviewer #1: My concern has been resolved.

Reviewer #2: The authors have addressed all my concerns, and I have no further questions.

Reviewer #3: It is revised well.

**Have all data underlying the figures and results presented in the manuscript been provided?**

Reviewer #1: Yes

Reviewer #2: Yes

Reviewer #3: None

PLOS authors have the option to publish the peer review history of their article (what does this mean?). If published, this will include your full peer review and any attached files.

Reviewer #1: **Yes: **Shuai Gao

Reviewer #2: **Yes: **Ji Dong

Reviewer #3: **Yes: **Quan Zou

**Data Deposition**

http://datadryad.org/submit?journalID=pgenetics&manu=PGENETICS-D-23-01385R1

**Press Queries**

---

## [Editor Report · Acceptance letter]

24 May 2024

PGENETICS-D-23-01385R1 

Single-cell and single-nucleus RNA-sequencing from paired normal-adenocarcinoma lung samples provide both common and discordant biological insights 

Dear Dr Renaut, 

We are pleased to inform you that your manuscript entitled "Single-cell and single-nucleus RNA-sequencing from paired normal-adenocarcinoma lung samples provide both common and discordant biological insights" has been formally accepted for publication in PLOS Genetics! Your manuscript is now with our production department and you will be notified of the publication date in due course.

With kind regards,

Olena Szabo

PLOS Genetics

On behalf of:
